# Bacillary Layer Detachment in Neovascular Age-Related Macular Degeneration: Case Series

**DOI:** 10.3390/biomedicines11030988

**Published:** 2023-03-22

**Authors:** Filomena Palmieri, Saad Younis, Walid Raslan, Lorenzo Fabozzi

**Affiliations:** 1Western Eye Hospital, Imperial College Healthcare NHS Trust, London NW1 5QH, UK; 2Surgery, Cancer and Cardiovascular Division, Ophthalmology Department, Imperial College Healthcare NHS Trust, London W2 1NY, UK; 3Moorfields Eye Hospital NHS Foundation Trust, London EC1V 2PD, UK

**Keywords:** bacillary layer detachment, age-related macular degeneration, neovascular AMD, anti-VEGF, retinal haemorrhage, myoid, macula, retina, intravitreal injections

## Abstract

Purpose: This study seeks to report the clinical and multimodal imaging findings of eight eyes of seven patients with neovascular age-related macular degeneration (nAMD) who developed bacillary layer detachment (BALAD). Setting/Venue: The patients were analysed at the Western Eye Hospital in London, UK. Methods: The approaches of this research include clinical examinations and multimodal imaging-based description of cases of nAMD with BALAD. Results: We report multimodal imaging findings of bacillary layer detachment (BALAD) in patients with nAMD. Conclusions: A bacillary layer detachment was detected in patients with neovascular age-related macular degeneration. This multimodal imaging finding is not commonly described in the literature for this disease.

## 1. Introduction

Bacillary layer detachment (BALAD) is a definition introduced for the first time in 2018 by Mehta et al. to describe an optical coherence tomography (OCT) finding in a patient with Toxoplasmosis chorioretinitis and pachychoroid disease [1]. It is characterized by a splitting of the photoreceptor layer at the myoid level, resulting in a space posterior to the external limiting membrane (ELM). Several authors have reported BALAD in various ophthalmic conditions as well as ocular trauma, multifocal placoid pigment epitheliopathy, pachychoroid syndrome, tubercular choroidal granuloma, acute idiopathic maculopathy, type 2 macular telangiectasia, Vogt–Koyanagi–Harada disease, choroidal metastasis, osteoma, and age-related macular degeneration (AMD) [1,2,3,4,5,6,7,8,9,10,11,12,13,14,15,16,17,18,19,20]. The exact BALAD physiopathology is unknown. However, inflammation, vascular changes, and choroidal thickening have been hypothesized as possible underlying mechanisms [2,5,7,11]. This study reports multimodal imaging findings of BALAD in patients with neovascular age-related macular degeneration (nAMD).

## 2. Methods

This study was conducted at Western Eye Hospital in London. Written informed consent was provided by all patients before participating in the study.

All the patients received a complete ophthalmic examination, including best corrected visual acuity (BCVA) with refraction using an Early Treatment Diabetic Retinopathy Study (ETDRS) chart, intraocular pressure (IOP) measurement measured via an iCARE tonometer, slit lamp evaluation, clinical examination of the posterior pole with a 66D or 90D indirect fundus-viewing lens, optical coherence tomography (OCT) and optical coherence tomography angiography (OCTA) of the macula region using the Spectralis HRA+OCT platform (Heidelberg Engineering, Heidelberg, Germany), and fundus autofluorescence (FAF) using Optos Ultra-widefield (Nikon Co., Ltd., Tokyo, Japan) or Spectralis HRA+OCT platform (Heidelberg Engineering, Heidelberg, Germany), at baseline. The medical history, ocular history, and a list of concomitant medications were collected before starting treatment.

The data used were obtained through Medisoft EMR software (Medisoft Limited, Leeds, UK).

At each follow-up visit, all patients had their BCVA and IOP measured, slit lamp evaluation, fundus oculi examination with a 66D or 90D indirect fundus-viewing lens, and OCT scan of the macular region.

## 3. Findings

### 3.1. Case 1

An 85-year-old male patient presented to our facility in April 2022, complaining of a left eye vision drop. The patient was on Timolol eye drops for glaucoma.

The BCVA was 0.1 and 1.0 LogMAR in the right and left eye, respectively. The anterior segment examination was unremarkable in both eyes.

Fundus examination showed macular drusen in the right eye and a macular yellowish elevated lesion with retinal haemorrhage in the left eye (Figure 1A). Autofluorescence imaging (FAF) showed a hypoautofluorescent macular lesion in the left eye (Figure 1B). An OCT scan was performed, showing a dry macula with drusen in the right eye and a type 2 macular neovascularization (MNV) with subretinal fluid and splitting at the ellipsoid zone (EZ) creating a hypo-reflective cavity in the left eye; a hyper-reflective band, continuous with the EZ layer, was observed on the floor of the BALAD (Figure 1C).

The optical coherence tomography angiography (OCTA) scan confirmed the presence of MNV in the left eye (Figure 2). Based on clinical findings and multimodal imaging, the patient was diagnosed with left nAMD. Anti-vascular endothelial growth factor (VEGF) intravitreal injections were commenced, according to a treat and extend (T&E) regimen. To date, the patient received four intravitreal injections. Since the first injection, the OCT scan showed a complete resolution of the BALAD (Figure 1D). However, a retinal pigment epithelium (RPE) tear was noted (Figure 1E). The BCVA did not improve (1.5 logMAR on the last examination).

### 3.2. Case 2

A 79-year-old male patient with a pigmented lesion along the vascular inferotemporal arcade presented to our care in August 2017 for the first time.

The anterior segment was within normal limits in both eyes.

The OCT scan showed a macular choroidal lesion with subretinal fluid in the right eye and macular drusen in the left eye. A right eye ultrasound scan was also performed, showing a macular choroidal nevus. His BCVA was 0.7 and 0.00 LogMAR in the right and left eye, respectively.

The right eye lesion was monitored for more than a year until the patient experienced a sudden right eye vision drop. His right eye BCVA was counting fingers (CF).

Fundus examination showed a right macular yellowish elevated lesion with retinal haemorrhages on the superior border of the naevus (Figure 3A). FAF showed a hypoautofluorescent macular lesion (Figure 3B).

An OCT scan was performed showing a type 1 MNV with a haemorrhagic BALAD on the superior border of the nevus. The BALAD cavitation showed hyper-reflective material. Hyper-reflective granular foci at the ceiling of the BALAD were reported. The ELM was visible anterior to the BALAD (Figure 3C).

The OCTA scan confirmed the presence of the right MNV.

Neovascular AMD was diagnosed. The patient started anti-VEGF intravitreal injections according to a T&E protocol. After four injections, the BALAD showed a partial response to the treatment (Figure 3D). The BCVA improved to 0.8 logMAR.

However, the patient decided to stop the treatment for personal reasons.

In the following 8 months, follow-up visits were performed. A progressive increase in intraretinal fluid, subretinal hyper-reflective material, and subretinal fluid were observed (Figure 3E).

The BCVA remained stable.

### 3.3. Case 3

A 62-year-old male patient presented to our care in December 2020 with a drop of vision in the left eye. His BCVA was 0.00 and 0.3 logMAR in the right and left eye, respectively.

The anterior segment examination was unremarkable in both eyes. An ophthalmological assessment showed features of right eye dry AMD and left eye nAMD with BALAD.

Fundus examination showed right macular drusen and a left macular subretinal fluid with a round yellow border (Figure 4A). FAF was also performed, showing hypoautofluorescence (Figure 4C).

The OCT scan showed a right dry macula with drusen and a left macular type 1 MNV with a fibrovascular pigment epithelial detachment (PED), subretinal fluid, and a split at the myoid zone creating an intraretinal space with multiple hyper-reflective foci and a septa-like arrangement. A hyper-reflective granular band at the anterior border of the BALAD was reported. The posterior border showed a hyper-reflective thickened band. The ELM was intact, recognizable, and anterior to the BALAD (Figure 4D,E).

The presence of the left MNV was confirmed by an OCTA scan.

An anti-VEGF intravitreal injection T&E regimen was commenced. However, the patient was non-compliant.

In March 2022, the patient decided to stop the treatment.

During the T&E treatment, the left eye developed subretinal fibrosis (Figure 4B) and no signs of visual acuity improvement were noticed in the following visits.

### 3.4. Case 4

An 82-year-old male patient presented to our care in June 2019 with a drop of vision in the right eye. His BCVA was 0.7 and 0.3 logMAR in the right and left eye, respectively.

The anterior segment examination was unremarkable in both eyes. An ophthalmological assessment showed features of right nAMD with BALAD and left vitreomacular traction.

Fundus examination showed a right macular yellowish slightly elevated lesion with subretinal haemorrhage (Figure 5A). FAF revealed a hypoautofluorescent macular lesion (Figure 5B).

The OCT scan showed a type 1 MNV with subretinal and intraretinal fluid and BALAD in the right eye. The BALAD cavity presented hyper-reflective granular foci (Figure 5C).

The left eye OCT scan showed vitreomacular traction.

The OCTA scan confirmed the presence of the right MNV.

An anti-VEGF intravitreal injection T&E regimen was commenced.

In October 2021, the patient could not continue the treatment due to hospitalization related to other health issues.

In April 2022, he presented to our care complaining of deterioration of vision in his right eye.

The BCVA was CF, and the anterior segment examination was unremarkable. The OCT scan showed worsening of the BALAD and nAMD features. Thus, the anti-VEGF intravitreal injections treatment was recommenced.

To date, the OCT scan showed an improvement in the subretinal and intraretinal fluid and a partial resolution of the BALAD. Hyper-reflective foci were present in the BALAD cavitation (Figure 5D).

However, no visual gain was observed.

### 3.5. Case 5

A 78-year-old female patient was referred to our department with nAMD in both eyes. She had a long history of different types of anti-VEGF intravitreal injections since 2012.

She showed features of BALAD in both eyes since the initial diagnosis was made. Her initial BCVA was 0.4 logMAR in both eyes.

The anterior segment examination was unremarkable in both eyes.

Fundus examination showed a macular yellowish elevated lesion in both eyes (Figure 6A,B). FAF revealed a hypoautofluorescent macular lesion (Figure 6C,D).

The OCT showed a type 1 MNV with a fibrovascular PED, subretinal fluid, and BALAD in both eyes (Figure 6E,F).

The OCTA scan confirmed the presence of MNV (Figure 7A,B).

Over the years, these features have shown improvement phases alternating to worsening ones.

The patient is still on treatment with anti-VEGF intravitreal injections (T&E regimen) for both eyes.

The BCVA has been stable throughout the treatment period. When last examined, it was 0.63 logMAR in the right eye and 0.32 logMAR in the left eye.

### 3.6. Case 6

A 79-year-old male patient presented to our facility in February 2019 complaining of vision distortion in the left eye. His BCVA was 0.2 logMAR in both eyes.

The anterior segment examination was unremarkable. An ophthalmological assessment showed features of right dry AMD and left nAMD with BALAD.

Fundus examination showed right macular drusen and a left macular yellowish elevated lesion with massive retinal haemorrhage.

FAF revealed a hypoautofluorescent macular lesion in the left eye (Figure 8).

The OCT scan showed a dry macula with drusen in the right eye and a type 1 MNV with PEDs, intraretinal and subretinal fluid, subretinal haemorrhage, and a haemorrhagic BALAD in the left eye (Figure 9A).

The OCTA scan confirmed the presence of the left MNV.

A T&E regimen with anti-VEGF intravitreal injection therapy was commenced. The patient showed an initial good response to the treatment (Figure 9B). Unfortunately, over the following years, the worsening of the condition was observed despite treatment. Thus, the patient switched to a different anti-VEGF molecule.

To date, the patient is still on treatment. The left BCVA was 0.4 logMAR when last examined. The last OCT scan showed intraretinal fluid and an improved BALAD (Figure 9C).

Figure 10 shows Optos Ultra-widefield colour and autofluorescence imaging of the left eye macular lesion two years after the start of the treatment.

### 3.7. Case 7

A 93-year-old male patient with a diagnosis of peripapillary choroidal neovascularization in the right eye, for which he has been receiving intravitreal injections for over 7 years, presented to our facility.

His BCVA was 0.5 and 0.1 logMAR in the right and left eye, respectively. The anterior segment examination was unremarkable in both eyes.

Fundus examination revealed a right greyish peripapillary lesion and macular drusen in the left eye. FAF was also performed, showing a right peripapillary hypoautofluorescent area (Figure 11).

The OCT scan demonstrated a type 1 MNV with PED and BALAD in the right eye. The ceiling of the BALAD presented hyper-reflective foci. The EZ zone was attenuated at the floor (Figure 12).

The OCTA scan confirmed the presence of the right peripapillary MNV (Figure 13).

These features were shown to be partially responsive to the treatment. The BVCA was stable over the treatment period.

## 4. Discussion

The term BALAD was recently introduced to indicate a splitting of the photoreceptor layer at the myoid level [1]. Assuming that the photoreceptors’ IS myoid is weaker than the ellipsoid and the junctional complexes of the ELM, it has been postulated that the main driving factor for developing BALAD could be the intrinsic weakness of the photoreceptors’ IS myoid [1,21,22]. When chorioretinal inflammation and exudation occur, the forces that promote the attachment of the photoreceptors’ outer segment (OS) to the RPE may be superior to the tensile strength of the photoreceptors’ IS myoid, resulting in splitting [1,21,22]. In nAMD, an IS photoreceptor shedding occurs due to degeneration. It may predispose nAMD patients to develop BALAD [23,24].

We examined the clinical and multimodal imaging findings of eight eyes of seven patients with neovascular age-related macular degeneration who developed bacillary layer detachment.

As reported in the literature, we had non-haemorrhagic BALAD in seven eyes and one haemorrhagic BALAD in one eye.

Ramtohul et al. found that there was a twofold risk of the development of fibrosis through 4 years in eyes presenting with haemorrhagic BALAD [12,20]. The haemorrhagic BALAD was reported in a case of MNV associated with macular telangiectasia type 2 [12]. It is important to differentiate between haemorrhagic BALAD and a submacular haemorrhage because the latter may require additional surgical treatment. As known in the literature, the resolution of macular haemorrhage in nAMD patients may result in a gradual transformation of the sub-RPE fluid into subretinal fibrosis and severe photoreceptor loss [20].

In our report, haemorrhagic BALAD showed a hyper-reflective cavity. These findings made the recognition of the retinal layers difficult. The patient showed an initial good anatomical response, but, over the following years, worsening of the condition was observed. The patient’s final BCVA was worse than the initial presentation due to the development of photoreceptor loss and subretinal fibrosis.

The non-haemorrhagic BALAD cavity showed hypo- or hyper-reflective material. In some cases, the hyper-reflective material may have a septa configuration. Some of our patients had a septa configuration of non-haemorrhagic BALAD. It is particularly evident in case reports 1 and 3.

It has been hypothesized that this material in the BALAD cavity consists of inflammatory elements, such as fibrin, and photoreceptor debris [1,21].

We postulated that some NVMs have an intense exudative and inflammatory component with consequent strong traction in the photoreceptor layer and formation of BALAD. It could be sustained by the inflammatory findings in the BALAD cavity.

We reported hyper-reflective granular foci at the ceiling of the BALAD. As reported in the literature, they could represent the myoid zone fragments and photoreceptors IS and OS. The ELM was visible superior to the BALAD on OCT scans.

We also observed a hyper-reflective band contiguous to an attenuated EZ on the floor of the BALAD. These findings have been reported in the literature [10,14,15,16,20,21]. Ramtohul et al. postulated that floor hyper-reflectivity could be related to the remaining mitochondria of the residual photoreceptor IS on the RPE–basal lamina–Bruch membrane complex. Alternatively, they hypothesized that the BALAD floor signal may be attenuated by the overlying fluid, the myoid fragments, and the OS attached to the RPE–basal lamina–Bruch membrane complex [21]. To date, a few studies have reported BALAD in neovascular AMD (nAMD). Thus, the incidence of BALAD in nAMD is not well defined. Sari Yordi et al. noticed that the bacillary detachment was present in 7.4% (6 of 81) of the eyes reviewed. It was predominantly associated with type 1 and 2 mixed MNV, higher fluid volumes, increased EZ attenuation, and sub-RPE disease [15]. Instead, Jae Hui Kim et al. reported an incidence of bacillary detachment in 4.5% (20 of 442) of the patients examined [16]. In a Korean cohort, Kim et al. observed that the incidence of BALAD was significantly different in the types of MNV. Type 2 MNV was found to be the most associated with BALAD [16]. These results accorded to the findings of Jung et al. [10]. Conversely, in the study by Ramtohul et al., type 1 MNV was reported to be the most associated with BALAD. They demonstrated BALAD in association with other MNV subtypes [20]. In our study, all cases presented a type 1 MNV, except for one patient who presented a type 2 MNV. Thus, our study is in agreement with the results reported by Ramtohul et al. The role of the presence of BALAD in the progression of nAMD is uncertain.

Ramtohul et al. hypothesized that the typical presence of extracellular matrix proteins in the subretinal space and the intra- and/or subretinal haemorrhages in BALAD may increase the risk of developing macular fibrosis in nAMD, after the fluid resorption. They found a similar distribution of subretinal fibrosis among the MNV subtypes, anti-VEGF agents, and treatment protocols [20]. We observed the development of macular fibrosis in two cases.

However, since anti-VEGF therapy in itself creates a profibrotic environment, at the moment it is not possible to define the role of BALAD in the development of fibrosis. Further studies are necessary.

In one of our cases, we noticed a spontaneous retinal pigment epithelial (RPE) tear in a type 2 MNV after four anti-VEGF intravitreal injections. To the best of our knowledge, there is no evidence of the relationship between BALAD and RPE rip development in the literature.

RPE tears have been reported as part of the natural history of nAMD. Risk factors in patients with AMD receiving anti-VEGF medications include larger lesion linear diameter, vertical PED dimension, and duration of PED formation. Bird proposed that RPE tears are a result of hydrostatic pressure from fluid or material accumulating in the sub-RPE space (in the PED) and eventually causing the rupture of RPE. However, there is a low incidence of retinal pigment epithelium tears after anti-VEGF intravitreal injections [25,26,27].

Mehta et al. suggested that a hydrostatic force from the choroid strong enough to split the photoreceptors was needed to form a BALAD [1].

Based on the probable assumptions of the BALAD formation and RPE tear development, we hypothesize that there may be some mutual mechanisms in the development of both. BALAD could be an additional risk factor for the early development of an RPE rip, but studies with longer follow-ups of these patients are needed to confirm this hypothesis.

Case series reported in the literature assumed that COVID-19 infection and/or vaccination could be a risk factor in developing BALAD, considering the increased systemic inflammatory response in some patients and infection/vaccination with possible signs of subclinical ocular inflammation. Raphaela M. Fuganti et al. speculated that the occurrence of BALAD and a large amount of fibrin accretion in their patient with central serous chorioretinopathy (CSC) could be related to the systemic inflammatory condition observed in patients with COVID-19 [11]. However, this additional risk factor could not be identified in our study.

Currently, it remains unclear whether BALAD increases the risk of macular atrophy and fibrosis in long-term follow-up. Furthermore, it is unknown if the patients with BALAD need a specific follow-up or a specific treatment regimen. The current treatment of nAMD with BALAD remains the anti-VEGF therapy. Despite this, new treatment modalities are being developed for the treatment of retinal pathologies. More recently, to overcome the drawbacks associated with injecting anti-VEGF, researchers have developed novel ocular pharmacological nanoformulations with multiple bioactive properties for enhanced treatment of age-related macular degeneration [28]. In conclusion, more studies with longer follow-ups are necessary for better defining physiopathology, epidemiology, and the actual role of the presence of BALAD in nAMD.

## Figures and Tables

**Figure 1 biomedicines-11-00988-f001:**
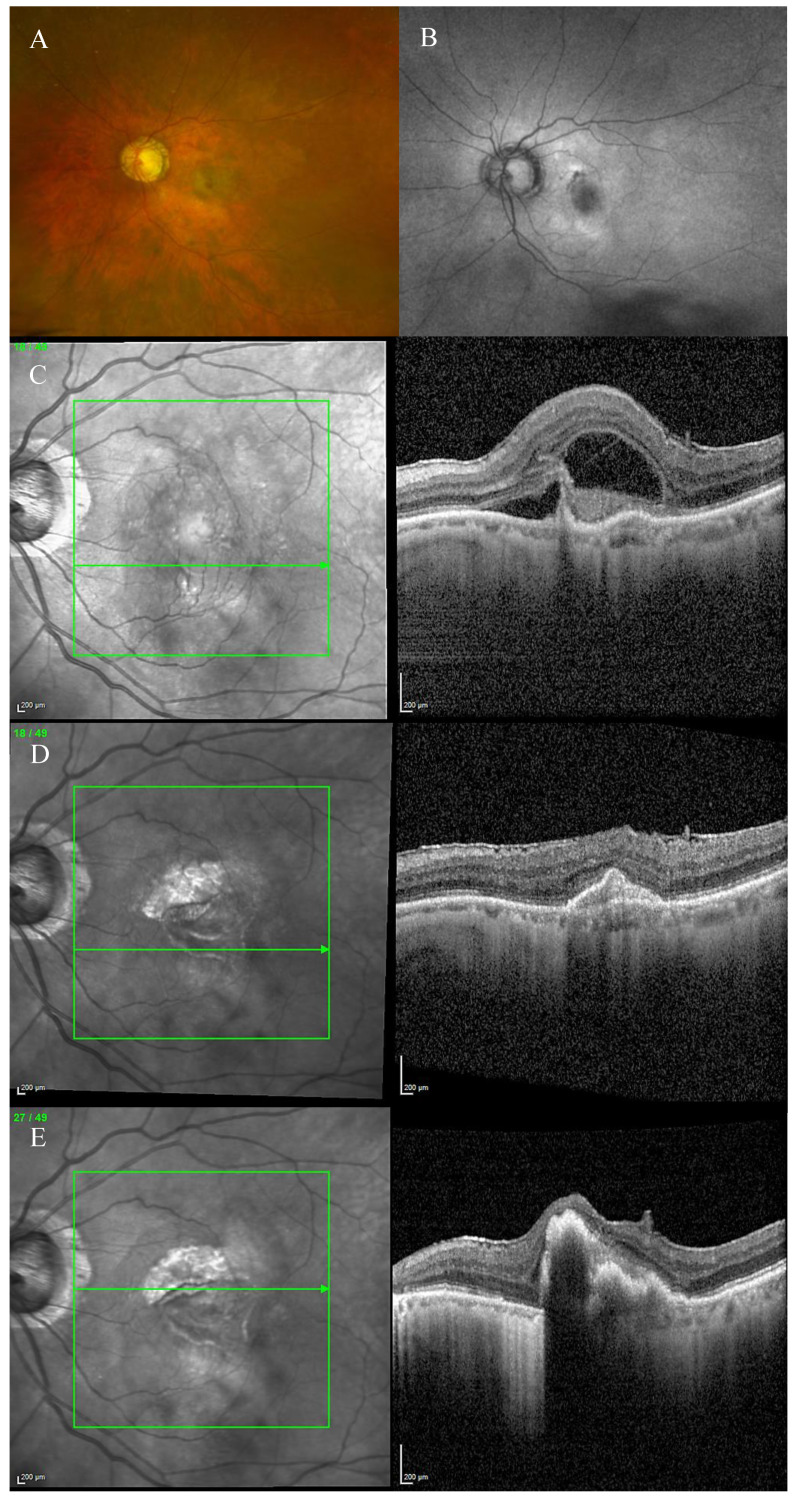
Case 1. Multimodal imaging features of bacillary layer detachment (BALAD) in a patient with left eye neovascular age-related macular degeneration. (**A**) Optos Ultra-widefield colour shows a macular yellowish elevated lesion with subretinal haemorrhage. (**B**) Optos Ultra-widefield autofluorescence imaging shows a hypoautofluorescent macular lesion. (**C**). *Heidelberg* Spectralis optical coherence tomography shows a type 2 macular neovascularization with splitting at the ellipsoid zone creating an hypo-reflective cavity and a hyper-reflective band on the floor of the BALAD. (**D**) Optical coherence tomography shows a complete resolution of BALAD after four anti-vascular endothelial growth factor intravitreal injections. (**E**) *Heidelberg* Spectralis optical coherence tomography shows a retinal pigment epithelial tear.

**Figure 2 biomedicines-11-00988-f002:**
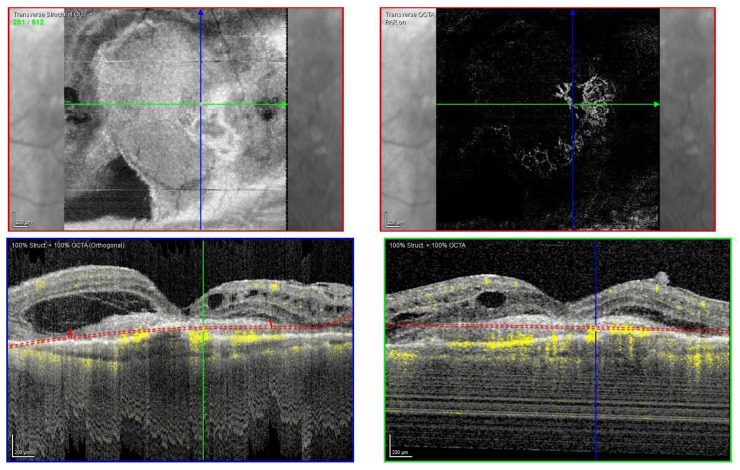
Case 1. Heidelberg Spectralis optical coherence tomography angiography shows a subretinal neovascular membrane in the left eye.

**Figure 3 biomedicines-11-00988-f003:**
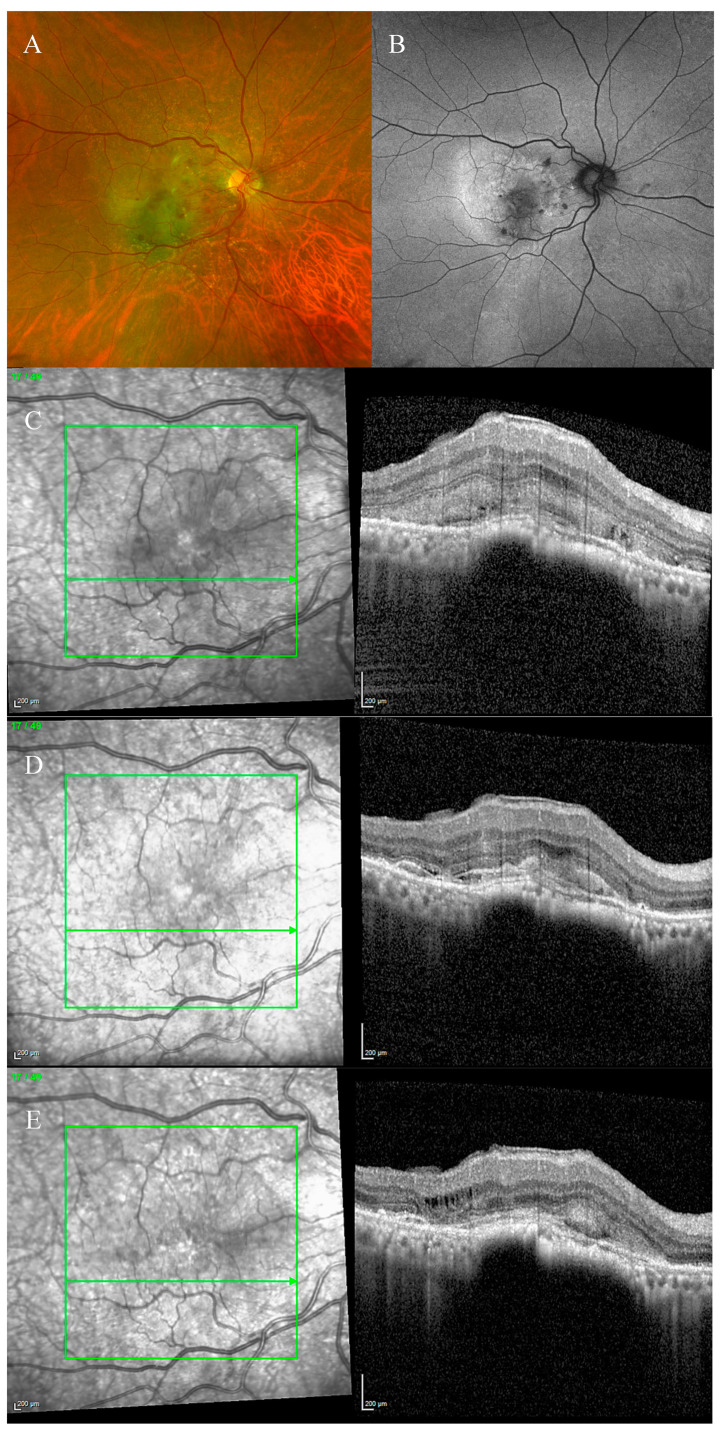
Case 2. Multimodal imaging features of bacillary layer detachment (BALAD) in a patient with right eye neovascular age-related macular degeneration on a choroidal naevus. (**A**) Optos Ultra-widefield colour shows a macular yellowish elevated lesion with intraretinal haemorrhages on the superior border of a pigmented lesion. (**B**) Optos Ultra-widefield autofluorescence imaging shows a hypoautofluorescent macular lesion. (**C**) Heidelberg Spectralis optical coherence tomography shows a type 1 macular neovascularization with BALAD. The BALAD cavitation showed hyper-reflective material. The external limiting membrane is visible anterior to the BALAD. (**D**) Heidelberg Spectralis optical coherence tomography shows an improvement in BALAD after 4 intravitreal injections. (**E**) Heidelberg Spectralis optical coherence tomography shows increased intraretinal fluid, subretinal hyper-reflective material, and subretinal fluid eight months after discontinuing the treatment.

**Figure 4 biomedicines-11-00988-f004:**
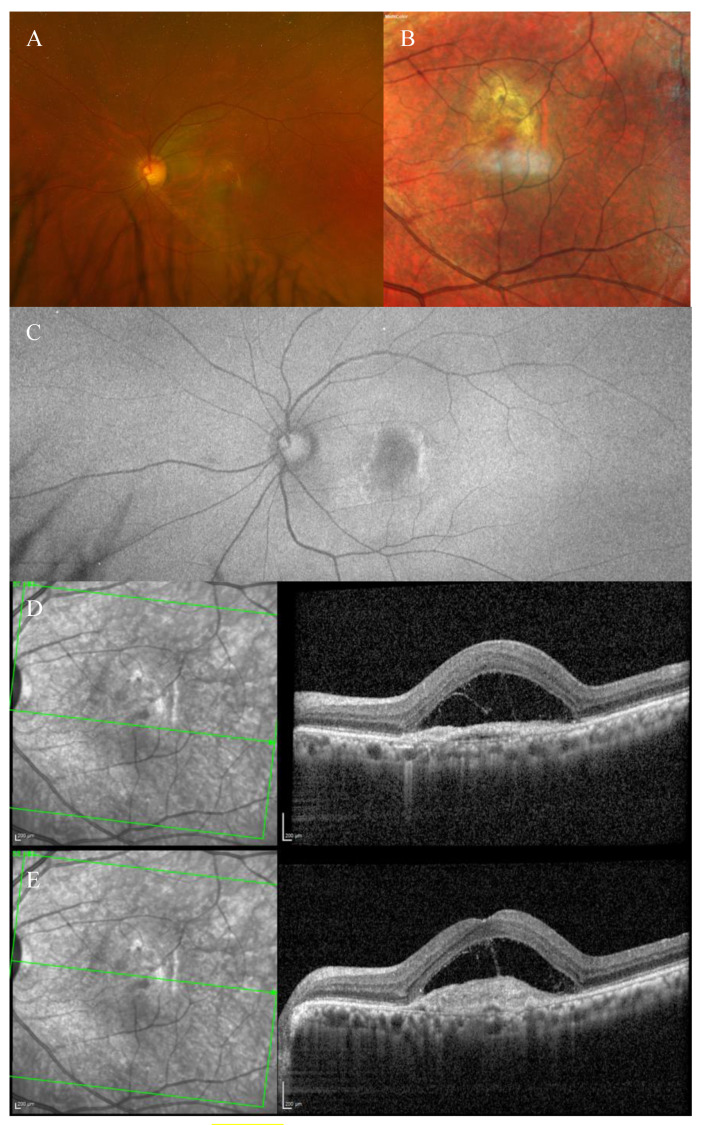
Case 3. Multimodal imaging features of bacillary layer detachment (BALAD) in a patient with left eye neovascular age-related macular degeneration. (**A**) Optos Ultra-widefield colour shows a macular subretinal fluid with a round yellow border. (**B**) Spectralis MultiColor shows a yellowish lesion with subretinal fibrosis. (**C**) Optos Ultra-widefield autofluorescence imaging shows a macular hypoautofluorescence lesion. (**D**,**E**) Heidelberg Spectralis optical coherence tomography shows a type 1 MNV with fibrovascular pigment epithelial detachment, subretinal fluid, and a split at the myoid zone creating an intraretinal space with multiple hyper-reflective foci and a septa-like arrangement and a hyper-reflective granular band at the anterior border of BALAD. The posterior border showed a hyper-reflective thickened band. The external limiting membrane is recognizable anterior to the BALAD.

**Figure 5 biomedicines-11-00988-f005:**
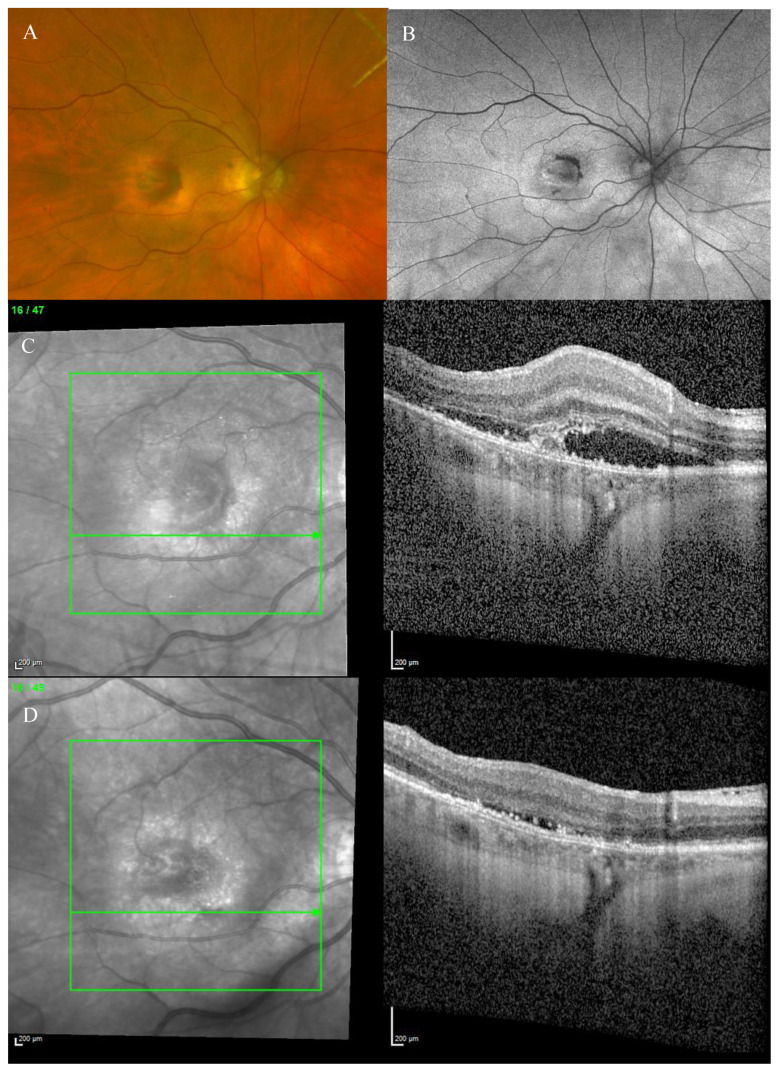
Case 4. Multimodal imaging features of bacillary layer detachment (BALAD) in a right eye neovascular age-related macular degeneration patient. (**A**) Optos Ultra-widefield colour shows a macular yellowish slightly elevated lesion with retinal haemorrhage. (**B**) Optos Ultra-widefield autofluorescence imaging shows a hypoautofluorescent macular lesion. (**C**) Heidelberg Spectralis optical coherence tomography shows a type 1 MNV with subretinal and intraretinal fluid and BALAD. (**D**) Heidelberg Spectralis optical coherence tomography shows a partial resolution of BALAD after anti-vascular endothelial growth factor intravitreal injections. Hyper-reflective foci are present in the BALAD cavitation.

**Figure 6 biomedicines-11-00988-f006:**
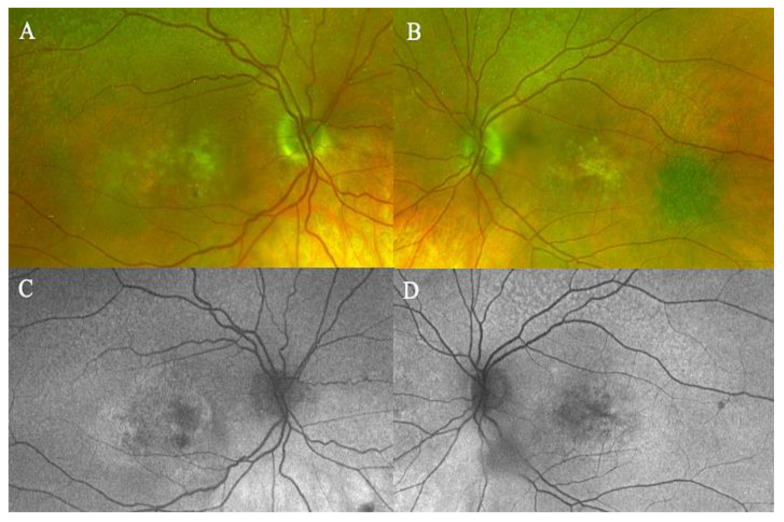
Case 5. Multimodal imaging features of bacillary layer detachment (BALAD) in neovascular age-related macular degeneration patients in both eyes. (**A**,**B**) Optos Ultra-widefield colour shows a macular yellowish slightly elevated macular lesion. (**C**,**D**): Optos Ultra-widefield autofluorescence imaging shows a hypoautofluorescent macular lesion. (**E**) Heidelberg Spectralis optical coherence tomography shows a type 1 MNV with subretinal and intraretinal fluid and BALAD in the right eye. (**F**) Heidelberg Spectralis optical coherence tomography shows a type 1 MNV with subretinal fluid and BALAD in the left eye.

**Figure 7 biomedicines-11-00988-f007:**
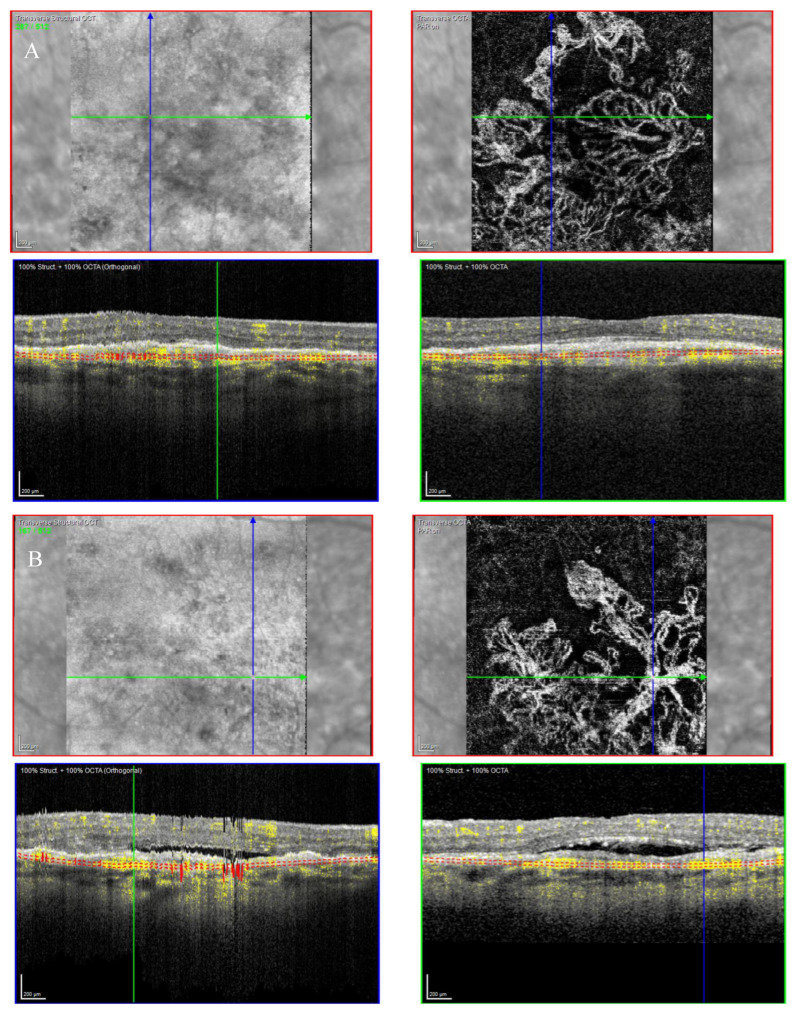
Case 5. Heidelberg Spectralis optical coherence tomography angiography shows a subretinal neovascular membrane in right (**A**) and left eyes (**B**).

**Figure 8 biomedicines-11-00988-f008:**
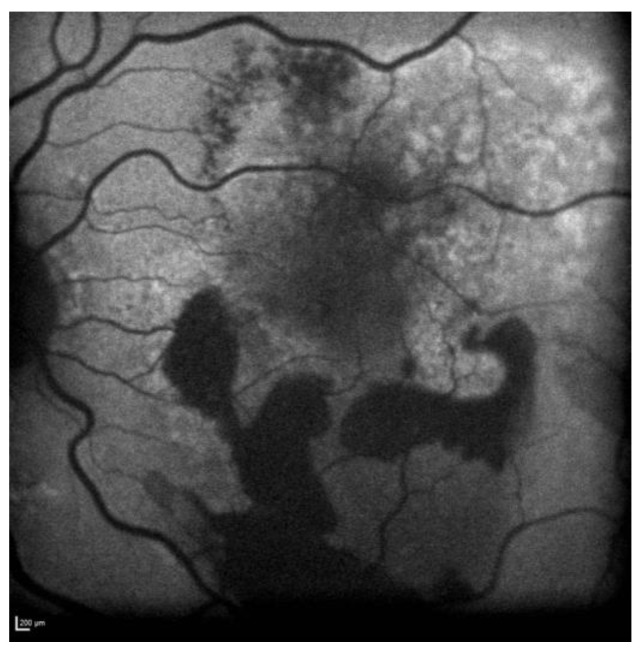
Case 6. Heidelberg Spectralis blue autofluorescence shows a left eye macular hypoautofluorescent lesion.

**Figure 9 biomedicines-11-00988-f009:**
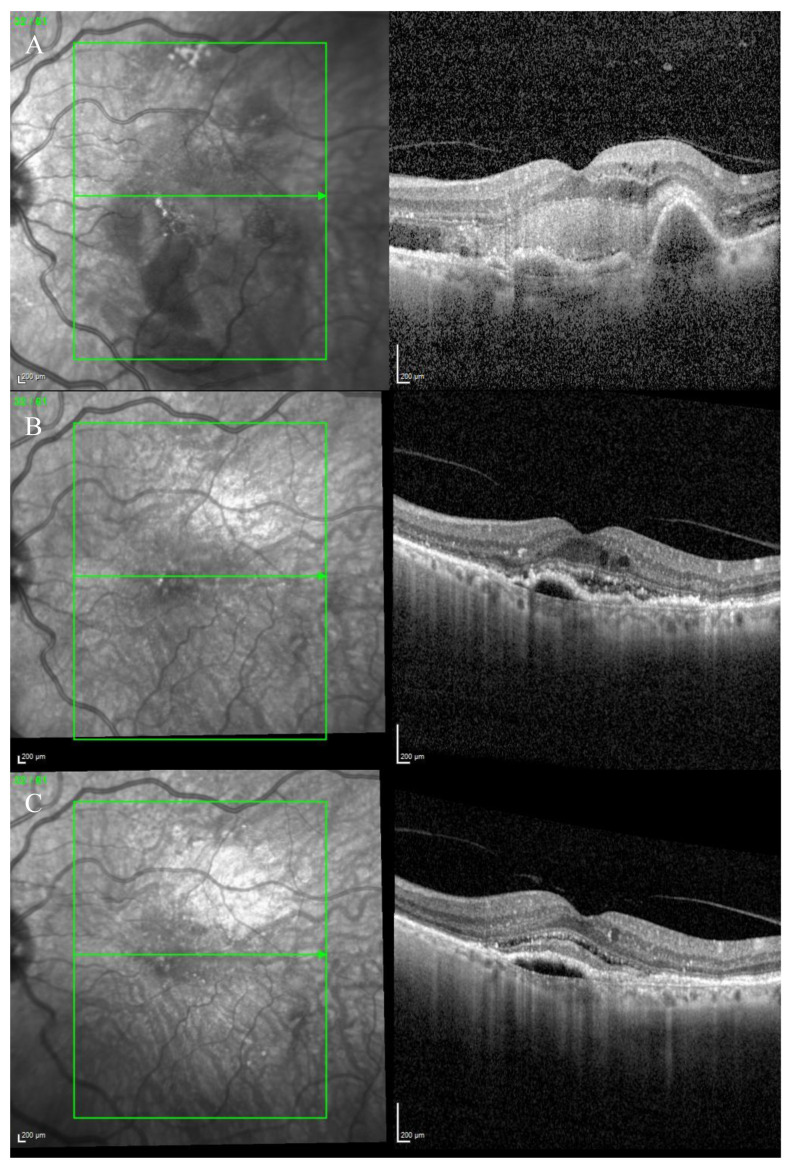
Case 6. (**A**) Heidelberg Spectralis optical coherence tomography shows a type 1 MNV with fibrovascular pigment epithelial detachment, subretinal and intraretinal fluid, and haemorrhagic BALAD in the left eye. (**B**,**C**) Heidelberg Spectralis optical coherence tomography shows a good response after a year and two years of intravitreal anti-VEGF treatment, respectively.

**Figure 10 biomedicines-11-00988-f010:**
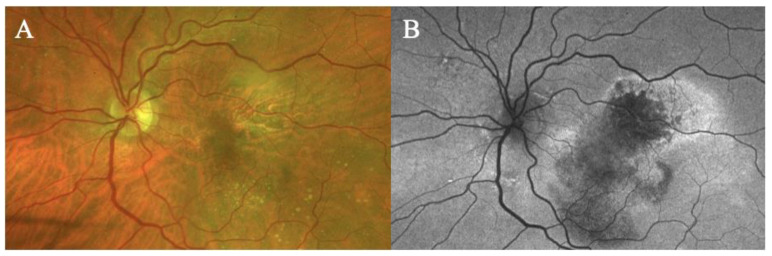
Case 6. Optos Ultra-widefield colour and autofluorescence imaging of the left eye after two years of intravitreal anti-VEGF treatment, showing a yellowish (**A**) and hypofluorescent (**B**) macular lesion. The retinal haemorrhage was fully resolved (**A**,**B**).

**Figure 11 biomedicines-11-00988-f011:**
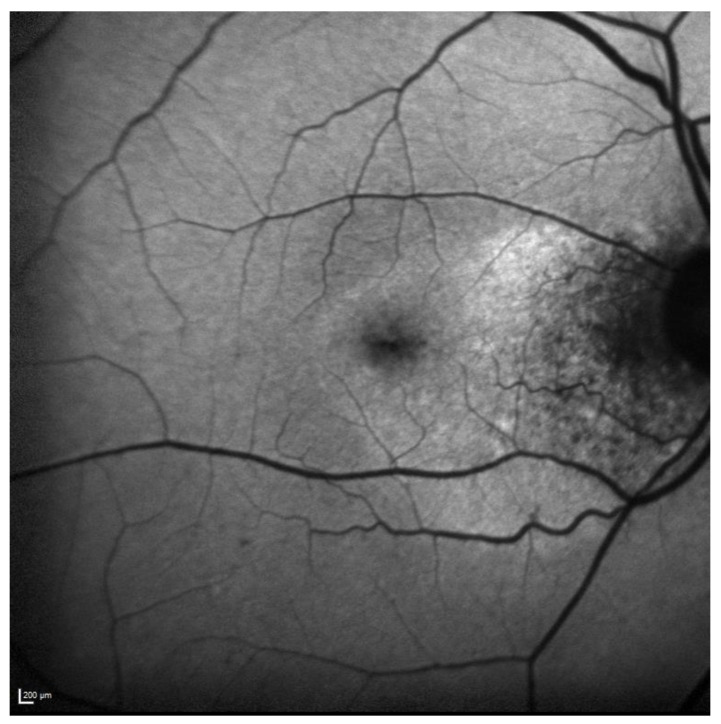
Case 7. Heidelberg Spectralis blue autofluorescence shows a peripapillary hypoautofluorescent lesion in the right eye.

**Figure 12 biomedicines-11-00988-f012:**
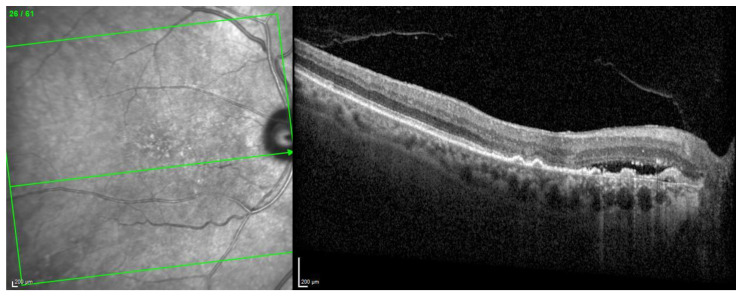
Case 7. Heidelberg Spectralis optical coherence tomography shows a peripapillary type 1 MNV with subretinal fluid and BALAD in the right eye.

**Figure 13 biomedicines-11-00988-f013:**
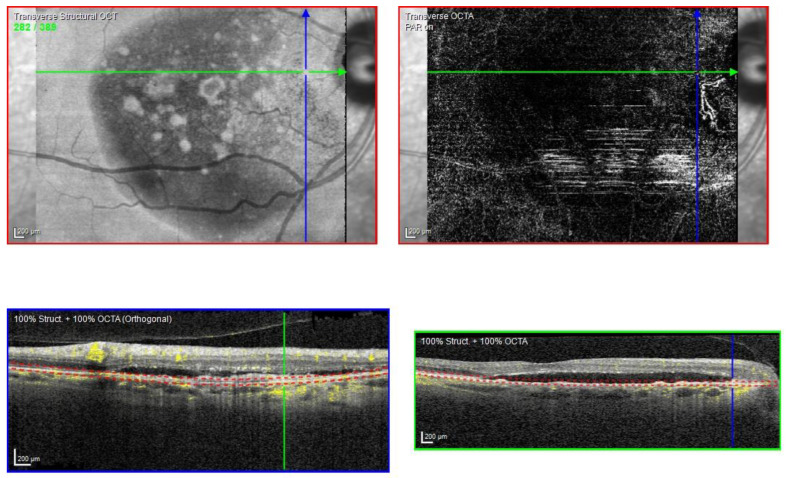
Case 7. Heidelberg Spectralis optical coherence tomography angiography shows a peripapillary subretinal neovascular membrane in the right eye.

## Data Availability

The data are not publicly available due to confidentially issues and will only be available on reasonable request from the corresponding authors.

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
