# Peer review of "Bacillary Layer Detachment in Neovascular Age-Related Macular Degeneration: Case Series"

_biomedicines, 2023, doi:10.3390/biomedicines11030988_

Round 1

Reviewer 1 Report

The current manuscript aims to report the clinical and multimodal imaging findings of 8 eyes of 7 patients with neovascular age-related macular degeneration who developed bacillary layer detachment. Although the case series are important to explore the bacillary layer detachment in neovascular age-related macular degeneration, the incidence, clinical features, and response to anti-VEGF therapy have been described before (please refer to the following papers: #1 DOI: 10.1016/j.oret.2022.05.022 & #2 DOI: 10.1016/j.oret.2021.09.010 & #3 DOI: 10.1097/IAE.0000000000003437). The authors should carefully clarify the differences in the academic contribution points between the current manuscript and those earlier reports. Furthermore, from the viewpoint of ocular clinical medicine, the exact physiopathology of bacillary layer detachment is unknown and raises the readers’ curiosity. The authors should describe pertinent information in detail. According to my checking, the authors indeed describe the patient’s history, diagnosis, and treatment. But, the audiences are curious about how to draw a conclusive summary of BALAD role in the progression of nAMD? In particular, each patient presented different case studies. The Discussion section should be better improved and deepened. In fact, many treatment modalities have been developed for the treatment of retinal diseases. More recently, to overcome the drawbacks associated with anti-VEGF injection, investigators have developed new ocular nanotherapeutics with multiple bioactive properties for potential use in the management of retinal disorders (DOI: 10.1021/acsnano.2c05824), which should be illustrated to balance the scientific viewpoint and attract more attention from audiences. Given that the current report aims to explore the role of biomedicines for ocular clinical medicine, the authors are highly recommended to consider the inclusion of this relevant paper in the reference list to broaden and deepen the Discussion section of the article content.

Author Response

Dear reviewer,

Thank you for your review and for your interesting suggestions.

My colleagues and I have made the changes requested.

We have improved the discussion, extended the references, reported more details of the BALAD in nAMD described in the literature, and improved our conclusion on the features and the role of BALAD.

Also, we have included the interesting article you suggested.

Thank you for your consideration and for your time.

We appreciate your help.

Yours sincerely, 

Miss Filomena Palmieri

Reviewer 2 Report

The paper entitled “Bacillary layer detachment in neovascular age-related macular degeneration: case series” is based on an interesting case series regarding bacillary layer detachment (BALAD) described by optical coherence tomography (OCT) results.

The results nicely present the characteristics of the patients with different forms of BALAD. Considering that this pathology has only been defined since 2018, studies reporting specific cases and OCT scans can help in the clinical setting when faced with patients with untypical scans. It would interesting if the authors could comment on whether or not COVID infection and/or vaccination, which appeared 2 years after the first description of BALAD, could be a risk factor in developing BALAD considering the increased systemic inflammatory response in some patients and infection/vaccination with possible signs of subclinical retinal/vitreal inflammation.

The paper is thorough and highlights the important issues and characteristics of patients with BALAD. The study adds to the literature and is of potential clinical interest. 

The authors should consider a flow chart on clinical diagnostics, management, and treatment of patients with acute loss of vision due to suspect BALAD based on OCT results, which could be helpful in a routine ophthalmologic setting.

The study has been correctly planned. It is nicely written and of clinical interest. References are appropriate. The figures and description of each case are interesting and assist in describing the results.

Author Response

Dear reviewer,

Thank you for your review and for your interesting suggestions.

My colleagues and I have made the changes requested.

We have improved the discussion, extended the references, reported more details of the BALAD in nAMD described in the literature, and improved our conclusion on the features and the role of BALAD.

Also, we have commented on the possible role of COVID in developing BALAD.

Thank you for your consideration and your time.

Yours sincerely,

Miss Filomena Palmieri

Round 2

Reviewer 1 Report

The revised version has adequately addressed the critiques raised by this reviewer and is now suitable for publication in "Biomedicines".